# Impacts of the Saharan air layer on the physical properties of the Atlantic tropical cyclone cloud systems: 2003-2019

Hao Luo[1], Yong Han[1,2,3*]

[1]Advanced Science & Technology of Atmospheric Physics Group (ASAG), School of Atmospheric Sciences, Sun Yat-sen University, Zhuhai 519082, China
[2]Key Laboratory of Tropical Atmosphere-Ocean System (Sun Yat-sen University), Ministry of Education, Zhuhai 519082, China
[3]Southern marine Science and Engineering Guangdong Laboratory, Zhuhai 519082, China

* *Correspondence to*: Yong Han (hany66@mail.sysu.edu.cn)

**Abstract.** It is generally known that the tropical cyclone (TC) cloud systems (TCCS) in the North Atlantic region frequently occur during boreal summer, while the Saharan dust outbreaks concurrently. The Sahara air layer (SAL), an elevated layer containing Saharan dry air and mineral dust, makes crucial impacts on the generation and evolution of TCs. However, the effects of SAL on the physical (macro and micro) characteristics of the Atlantic TCCS have not been well constrained, and the interaction mechanisms between them still need further investigation. In this study, our primary interest is to distinguish the various effects of SAL on different intensities of TCs, and further find out the probable causes of the varied feedback mechanisms. Therefore, we attempt to identify whether and how the effects of the SAL play a positive or negative role on the TCCS, and to draw a qualitative conclusion of how SAL affects the various intensities of the TCs. This paper focuses on the 70 TC samples from July to September in the years of 2003-2019 to investigate the physical effects of SAL on three intensities of TCs, i.e.: the tropical depression (TD), tropical storm (TS), and hurricane (HU). The results show that SAL has a positive impact on the macro properties of HU but significantly suppresses the TD. It appears that the SAL attributes little to the variation of ice cloud effective radius ($CER_i$) for TS, whereas $CER_i$ changes significantly and differentially for TD and HU. When affected by SAL, the probability density function (PDF) curve of $CER_i$ generally shifts to the smaller value for TD, whereas the PDF curve becomes flatter for HU. Our analysis indicates that the various responses of TCCS to SAL are determined by the combined effects of dry air masses, the dust aerosols as ice nuclei, as well as the thermodynamic and moisture conditions. Based on the observation data analysis, a concept scheme description has been concluded to deepen our recognition of the effects of SAL on the TCCS.

## 1 Introduction

Tropical cyclones (TCs) at landfall are risky natural disasters that endanger human life and property (Zhou et al., 2018; Parks et al., 2021). Accurate TC forecasts and data analysis are essential for the protection of human beings (Gray, 1968; Sun and Zhao, 2020). In the past several decades, a series of advances in remote sensing observations, numerical simulations,

and data assimilations have improved our recognition of TC cloud systems (TCCS) microphysics, dynamics and thermodynamics (Ooyama, 1969; Dunion and Velden, 2004; Holt et al., 2015). The intensity forecasts still remain uncertain, though TC track forecasts have been markedly improved (Chen et al., 2010).

The North Atlantic is one of the areas with the most frequent TCs in the world, and these TCCS are often affected by the Saharan air layer (SAL), which is an elevated layer of Saharan dry air and mineral dust, especially during the boreal summer between July and September (Dunion and Velden, 2004). Saharan dust usually propagates downstream along with SAL to the Atlantic region, which can modify the Earth-atmosphere system energy budget through direct, semi-direct and indirect radiative forcing, respectively (Twomey, 1974; Carlson and Benjamin, 1980; Albrecht, 1989; Sassen et al., 2003; Garrett and Zhao, 2006; Zhao and Garrett, 2015; Luo et al., 2019). Saharan dust represents one of the main contributors to the primary aerosol load in the atmosphere (Gutleben et al., 2019). The suspended dust particles directly scatter the incident solar radiation, potentially increasing the amplitude of the easterly waves (Jones et al., 2004; Dunkerton et al., 2009) and influencing the formation and intensity of the North Atlantic TCs (Evan et al., 2006; Wu, 2007). In addition, mineral dust particles heat the atmosphere and further enhance cloud evaporation, referred to as the semi-direct effect. (Amiri-Farahani et al., 2017; Tsikerdekis et al., 2019; Luo et al., 2021). In microphysics, dust aerosols act as cloud condensation nuclei and ice nuclei, altering the cloud microphysics and optical properties (Lohmann and Feichter, 2005), and also affecting cloud lifetimes (Huang et al., 2019). Given that the cloud itself is a significant regulator of the Earth's radiation budget, the dust indirect effect thereby plays an important role in the radiative forcing. The extended cloud lifetime delays precipitation, which allows more small cloud particles to ascend above the freezing level and initial ice process, hence release more latent heat and invigorate the vertical convection of clouds (Rosenfeld et al., 2008). The climate impact of this invigoration effect is likely a positive forcing (Koren et al., 2010; Huang et al., 2014). Besides the dust aerosol effects, the entrainment of dry and stable air inhibits the occurrence of deep convection that is essential for TC formation (Dunion and Velden, 2004), and also affects cloud microphysics (Xu et al., 2021; Zhu et al., 2021).

At present, how the SAL affects TCCS is still a controversial issue in the TC research community. According to the previous studies, it seems that the SAL plays both a positive and negative role in the TC activities since different aspects of the impacts are previously emphasized. For instance, Karyampudi and Carlson (1988) and Karyampudi and Pierce (2002) concluded that the SAL can invigorate the easterly wave growth and TC development from the analysis of synoptic, vorticity budget, and positive vorticity gradient. But inversely, Dunion and Velden (2004) have demonstrated that the outbreaks of SAL suppress Atlantic TC activities from satellite observations, owing to the facts that the SAL enhances the convectively driven downdrafts, the wind shear, as well as the preexisting trade wind inversion, and thereby stabilizes the environment. Wong and Dessler (2005) and Evan et al. (2006) also suggested that there exists a strong negative relation between Saharan dust and TC activity. Sun and Zhao (2020) have conducted a comprehensive investigation into the effects of dust aerosols on the meteorological environment, including the vorticity, temperature, wind shear, and humidity, which indicated a negative effect on TC formation. Currently, the impacts of SAL on TC property is a scientific issue of debate, which needs further study.

According to the aforementioned efforts, few of them have conducted observational studies on how SAL affects the TCCS macro and micro properties for a fixed TC intensity from long-term measurements. Our goal of this study is to identify whether and how the effects of the SAL play a positive or negative role on the TCs, and to give a qualitative conclusion of how SAL affects the different intensities of TCs. Note that the meteorological parameters include sea surface temperature, vorticity, wind shear, specific humidity, and planetary boundary layer are all the factors for the formation and development of TCs (Hill and Lackmann, 2009; Wingo and Cecil, 2010; Tory and Dare, 2015; Han et al., 2019; Sun and Zhao, 2020). To limit the impacts of meteorological factors for the TC activities, this study comprehensively examines the feedback mechanisms of TCs to dry and dusty SAL under the same TC intensity, for instance, tropical depression (TD), tropical storm (TS) and hurricane (HU). Aerosol optical depth (AOD), cloud top temperature (CTT), cloud ice water path (CIWP) and ice cloud effective radius (CER$_i$) retrieved from the Moderate Resolution Imaging Spectroradiometer (MODIS) on the Aqua satellite are used to analyze the connection between SAL and TCCS over the tropical North Atlantic. Based on the 17 years (2003-2019) long-term observations, the variations of TCCS macro and micro characteristics with the outbreak of SAL are explored, which will be beneficial to further understand the potential effects of Saharan dust and key meteorological factors on the TC evolutions on large scales in the Atlantic region. Here, the TCCS macro characteristics include the CTT and CIWP, while the TCCS micro characteristics refer to the CER$_i$.

The paper is organized as follows. Section 2 describes the data and methods used in this study. Section 3 shows the results and discussion. The conclusions and summary are provided in section 4.

## 2 Data and Methods

To illustrate the variation characteristics of Atlantic TCCS influenced by the SAL, we use the aerosol and cloud data from MODIS-Aqua, the TC track data from NOAA's Tropical Prediction Center, and the meteorological data from the National Centers for Environmental Prediction-Final Operational Global Analysis (NCEP-FNL) dataset. Given the time when both TC and SAL are prevalent, the study period is from July to September during 2003-2019. More details are provided in the subsections below.

### 2.1 Study area

The 70 TC samples are selected at 12:00 UTC from July to September during 2003-2019, and their centers are located in the inner rectangle shown in Fig. 1. The uniformity of the TC spatial distributions ensures the reliability of the statistical results. The average TC radius is about 500 km (Knaff et al., 2014), therefore, it is believable that all the TC samples are covered in the outer rectangle (5 degrees outward from the inner rectangle) on the main path of the SAL. Since the clean TCCS from the south can interact with the polluted SAL over the Atlantic, the research area selected in this paper is ideal for studying the interaction between SAL and Atlantic TCCS and investigating the mechanism of aerosol-cloud interaction. The

intent of this work is to identify whether and how the effects of the SAL play a positive or negative role on the TCCS in the study area, and to draw a qualitative conclusion of how SAL affects the various intensities of the TCs.

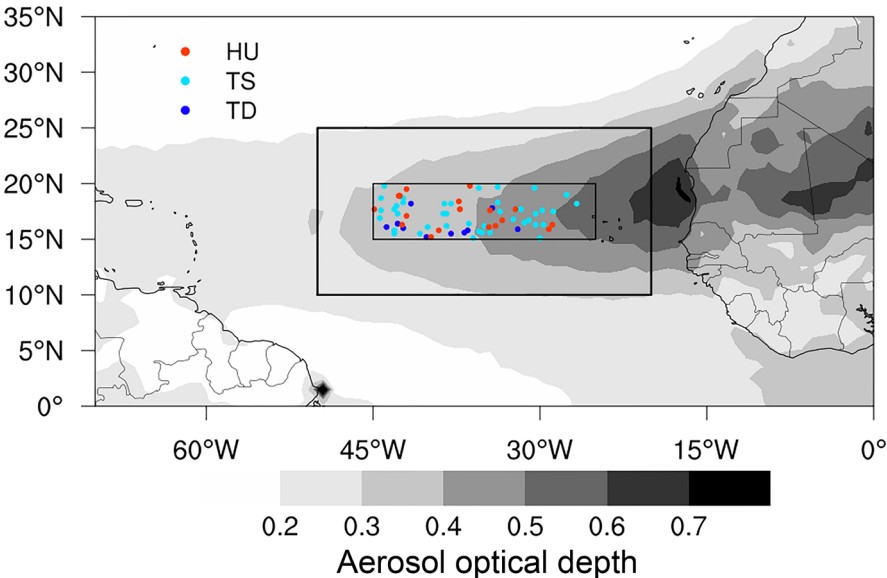

**Figure 1: The locations of the TC centers for different TC intensities, and the blue, green, and red dots represent the centers of TD, TS, and HU, respectively. The contour map is the spatial distribution of the average AOD from July to September during 2003-2019. The spatial ranges of the inner and the outer rectangles are [15° N-20° N, 45° W-25° W] and [10° N-25° N, 50° W-20° W], respectively. It should be noted that the AOD is generally averaged in the absence of clouds, i.e. no TC activities.**

## 2.2 Aerosol and cloud data

Aerosol and cloud data from July to September during 2003-2019 are taken from the MODIS-Aqua, which passes over the study area at about 13:30 local time. The version of the data is C6.1. To study the statistical properties of TCs under
pristine and polluted conditions, the MYD08-D3 daily gridded products at a spatial resolution of $1°×1°$ are used. The data include the AOD, CTT, $CER_i$, CIWP, etc.

Fig. 2 shows the probability density function (PDF) of the average AOD within ±5 degrees latitude or longitude away from each TC center. The PDF is in line with a normal distribution, and its peak value occurs when the ambient AOD is around 0.25 (mean value = 0.26 and median value = 0.25). According to this PDF of AOD, the value of 0.25 is taken into
account as the threshold when dividing the samples into pristine and polluted conditions. When AOD is less (greater) than 0.25, the ambient condition of the TC tends to be pristine (polluted). Previous studies suggest that the value of AOD is associated with the SAL intensity (Marenco et al., 2018; Huang et al., 2020), and thus the polluted or pristine conditions indicate whether or not the SAL outbreaks.

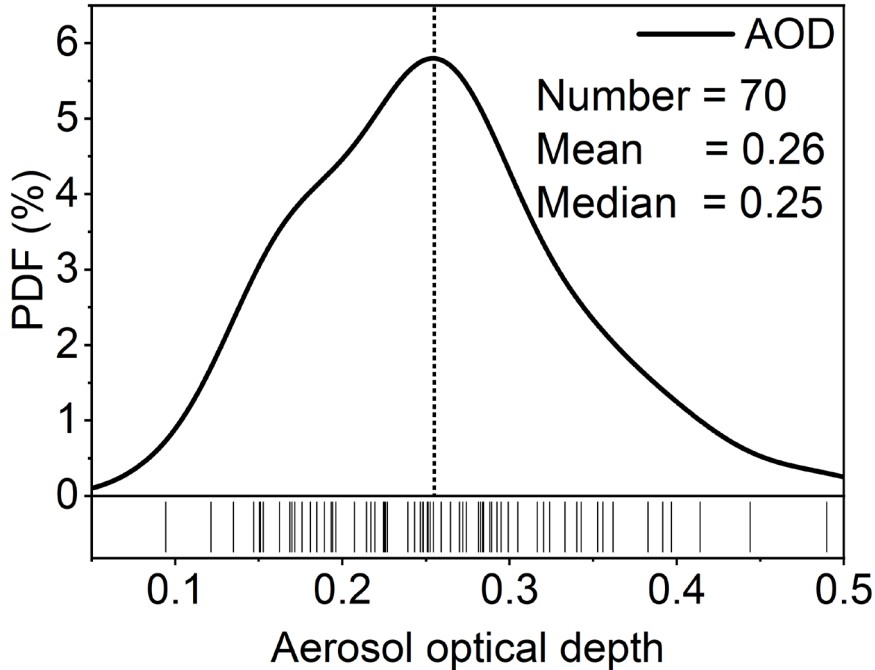

     **Figure 2: The PDF of the average AOD within ±5 degrees latitude or longitude away from each TC center. The dashed line indicates the peak perpendicular line of the PDF.**

## 2.3 Atlantic TC track data and meteorological data

Atlantic TC track data from July to September during 2003-2019 are obtained from NOAA's Tropical Prediction Center (https://www.nhc.noaa.gov/data/#hurdat). The TC track data are recorded at six-hour intervals with various TC
intensities, and the locations of the TC centers at 12:00 UTC are used in this study. According to the recorded data, 70 TC samples with three intensity categories are selected, including TD, TS, and HU. The locations of the TC centers are presented in Fig. 1, and the number of samples for each TC intensity under different polluted conditions is listed in Table 1.

**Table 1: The number of samples with different TC intensities under pristine and polluted conditions, as well as their total sample numbers.**

| TC intensity | Pristine | Polluted | Total |
|:---:|:---:|:---:|:---:|
| TD | 6 | 4 | 10 |
| TS | 21 | 21 | 42 |
| HU | 7 | 11 | 18 |

The meteorological data including temperature (T), dewpoint temperature ($T_d$), wind speed (WS), wind direction (WD), and relative humidity (RH) are taken from the NCEP-FNL global reanalysis dataset, which is produced by the Global Data Assimilation System (GDAS). The 1° × 1° gridded data at 12:00 UTC during the occurrence periods of the 70 TCs are used.

The water vapor flux ($F_{wv}$) is calculated from the FNL reanalysis data on each isobar (Liu et al., 2020). The FNL reanalysis datasets incorporate additional observations that are not available for inclusion in the real-time NCEP Global Forecast System (GFS) analysis, which have been widely applied to many studies (Ritchie et al., 2011; Kerns and Chen, 2013).

## 2.4 Methodology for data matching

Considering that this work involves multiple satellite and reanalysis data, such as AOD, CTT, CERi, and CIWP, Atlantic TC track data, meteorological data, etc., we provide more information about the data matching here. The specific study area is different for each TC, and each TC study area is located within ±5 degrees latitude or longitude away from the TC center, which is a grid of 11°×11°. The spatial resolution of all the data are 1°×1°, which allows the data to match well in the space. The time interval among the datasets is within 1.5 hours to guarantee a less changed atmospheric state. The CTT, $CER_i$ and CIWP in the study area are used to analyze the spatial TCCS properties. The AOD in the study area is averaged to characterize the SAL intensity. It should be noted that AOD is detected in the absence of clouds, so the AOD data are distributed on the periphery of the 11°×11° grid. Nevertheless, the properties of background aerosols are capable to be characterized by the average AOD. The meteorological data (T, Td, WS, WD, RH, etc.) in the study area are applied to examine the meteorological fields and profiles.

## 3 Results and Discussion

In this section, we discuss the impacts of SAL on the ice clouds of the Atlantic TCCS through the dust semi-direct and indirect effects, as well as the evaporation effect by the dry air masses in the SAL. The variations in physical properties of the TCCS with different intensities (TD, TS, and HU) caused by the SAL are analyzed due to their various dynamic-thermodynamic properties. In section 3.1, the general trajectories and structures of different intensities of TCs are summarized. Section 3.2 presents the background meteorological parameters. The detailed discussions of the impacts of SAL on the TCCS are provided in section 3.3. The relationship between sections 3.1-3.3 is demonstrated in Fig. 3.

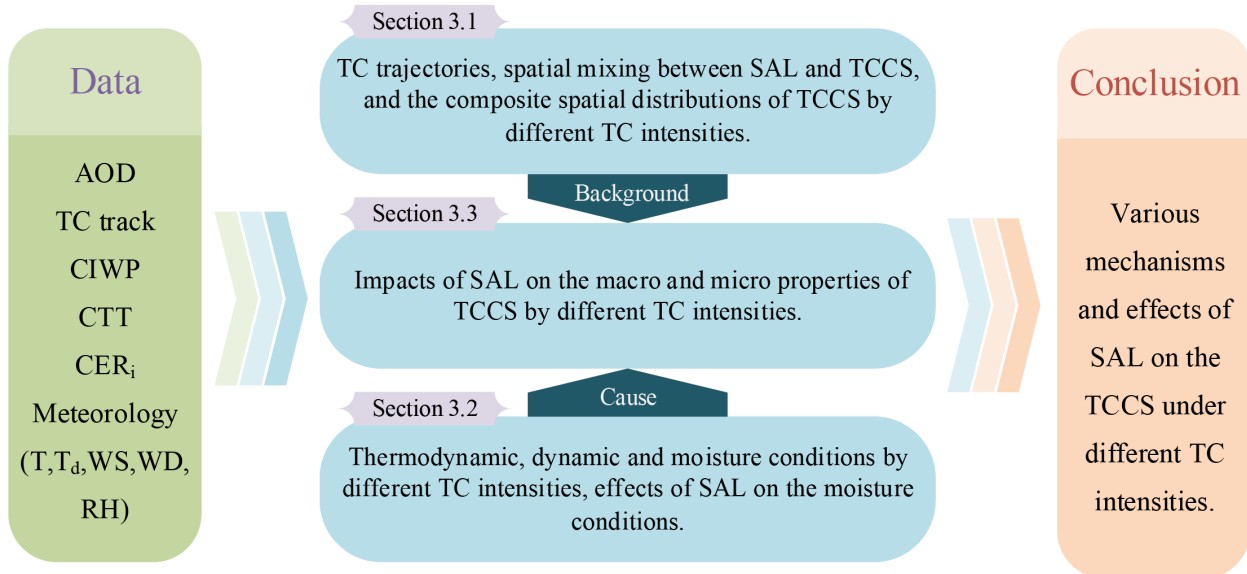

**Figure 3: The schematic diagram reflecting the relationship between sections 3.1-3.3.**

## 3.1 The overall characteristics of the TCs

The Atlantic TCs occur frequently from July to September, and during this period, the observed dust AOD over the North Atlantic reaches its peak (Xian et al., 2020). The dry and dusty air masses in the SAL originate from the west coast of Africa and then extend westward to the tropical Atlantic (Wang and Liu, 2014). At the same time, the Atlantic TCCS are influenced by the SAL, and the mixture of dry and dusty air masses with the TCCS will modify the macro and micro characteristics of the ice clouds (Nowottnick et al., 2018; Pan et al., 2018; Price et al., 2018).

Noting that adequate mixing spatially and temporally seems to affect the effectiveness of interactions between the SAL and the TCCS. Fig. 4 presents the tracks of the studied TD, TS, and HU under polluted conditions. The trajectories of these TCs are all from southeast to northwest guided by the ambient airflow and are located in the main area of the SAL. The Saharan dust and TCs both move from the coastline to the tropical Atlantic and interacts during the transport processes. Therefore, the trajectories ensure the mixtures of SAL with the TCs are sufficient.

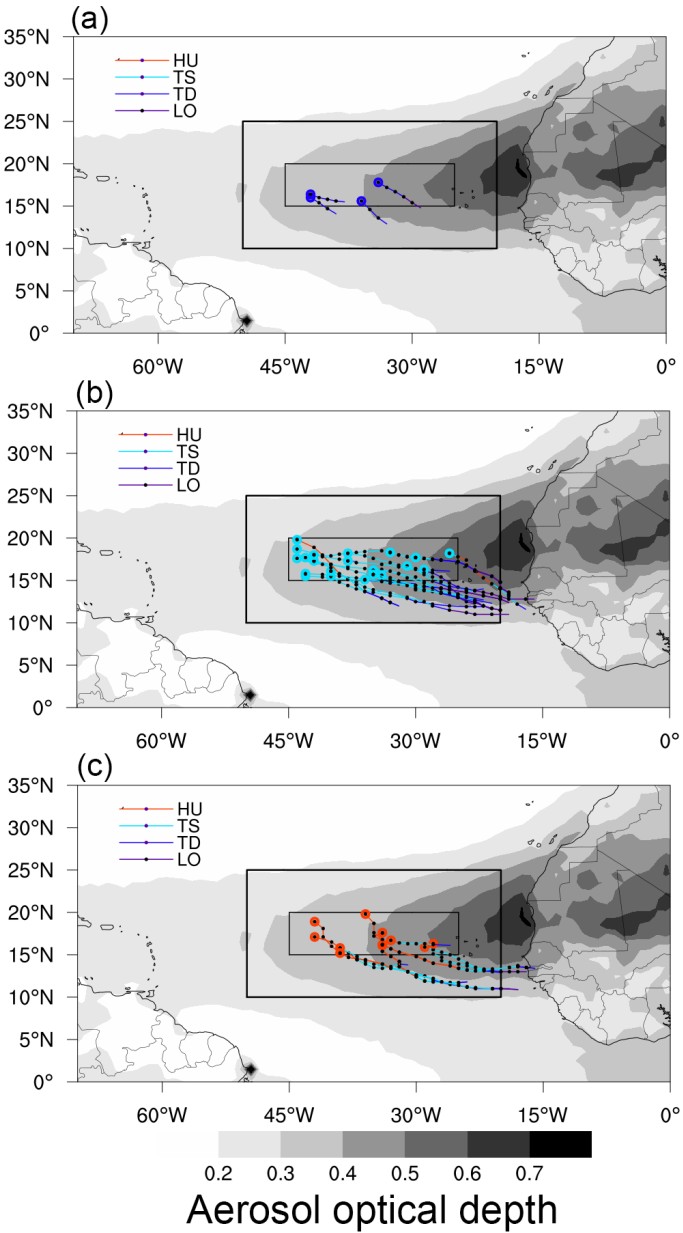

**Figure 4: The tracks of the TCs: (a) TD, (b) TS, (c) HU under polluted conditions. The blue, green, and red dots represent the centers of the studied TD, TS, and HU, respectively. The black dots are the TC centers every six hours before. The contour map is the spatial distribution of the average AOD from July to September during 2003-2019.**

The composite spatial distributions of CIWP and CTT under different TC intensities are shown in Fig. 5. When TC intensity increases, the CIWP increases and the CTT decreases. The high CIWP for HU is located on the TC center with its value greater than 0.8 kg m$^{-2}$. However, the high values of CIWP for TD and TS are only 0.5-0.8 g m$^{-2}$. The CTT of the TC

center is colder than 220 K for HU, between 220 K and 230 K for TS, and warmer than 230 K for TD. More water vapors are carried to the higher altitude by the stronger updraft in the HU, so the CIWP is higher and the CTT is colder. Moreover, with the increase of TC intensity, the cyclonic circulation becomes more organized. Stronger TCs generally have clear structures, and their intensities decrease uniformly outward from the centers (Emanuel, 2004). It should be noted that the spatial resolution of the data is 1° × 1°, which exceeds the average TC eye radius of 26 km (Knapp et al., 2018), so the TC eyes are not depicted in the composite spatial distributions.

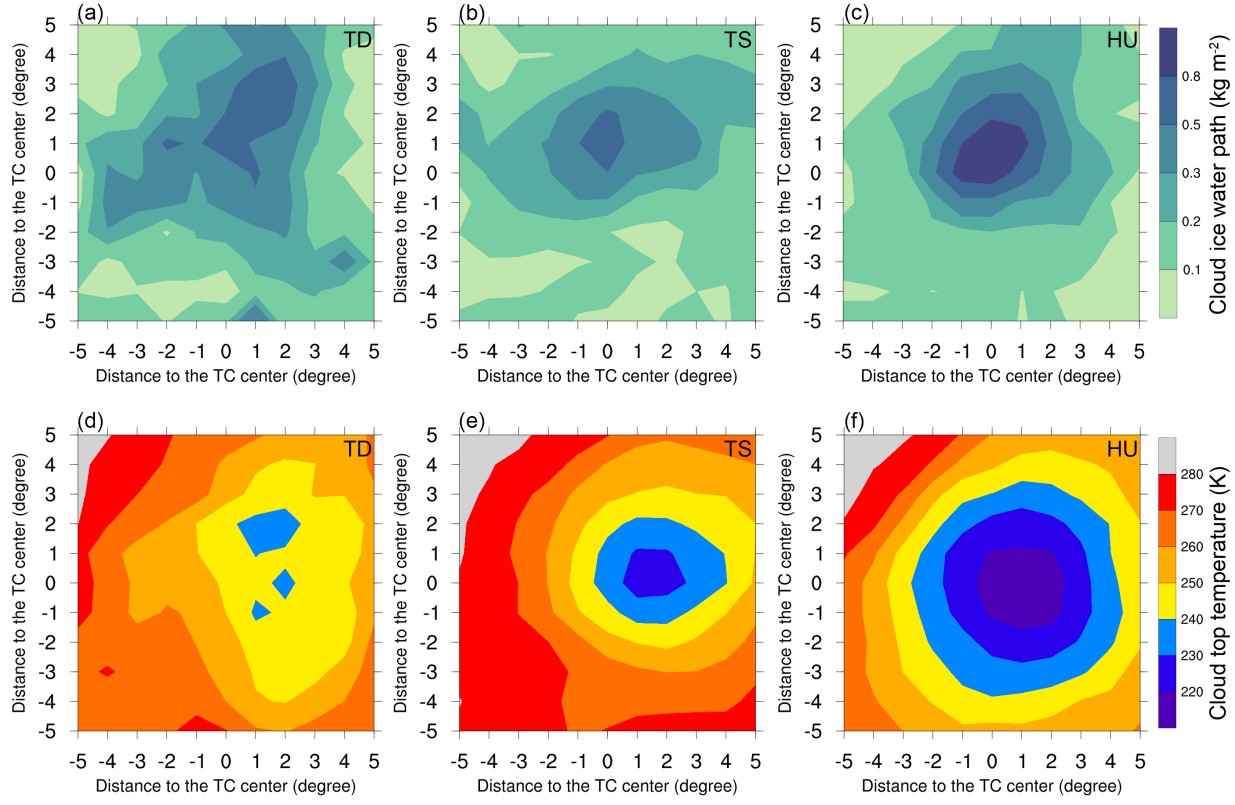

Figure 5: The spatial distributions of (a-c) CIWP and (d-e) CTT of the composite TD, TS, and HU, respectively.

## 3.2 Meteorological fields and profiles

TCCS are fueled by water vapor, and moist convection is the primary driving force for TC formation and intensification (Montgomery and Farrell, 1993). Theoretical studies have viewed that TCs are closed steady-state thermodynamic cycles, receiving heat and moisture from the ocean (Emanuel, 1986). Oceanic evaporation within the TC area is considered to increase with the wind speed (Makarieva et al., 2017).

Fig. 6 depicts the water vapor flux and wind field on the isobaric surfaces of 500 hPa, 700 hPa, and 850 hPa, respectively. It is clear that the wind fields of TD and TS are influenced by the easterly trade wind, and the convergence of the cyclone itself is not obvious. Strong wind speed and convergence are the significant characteristics of the intense TCs,

which supply sufficient water vapor from the ocean to HU. Likewise, Ermakov et al. (2015, 2019) suggested a positive correlation between TC intensification and atmospheric moisture supply. Therefore, the differences in water vapor supply caused by dynamic conditions are largely determined by the intensities of TCs.

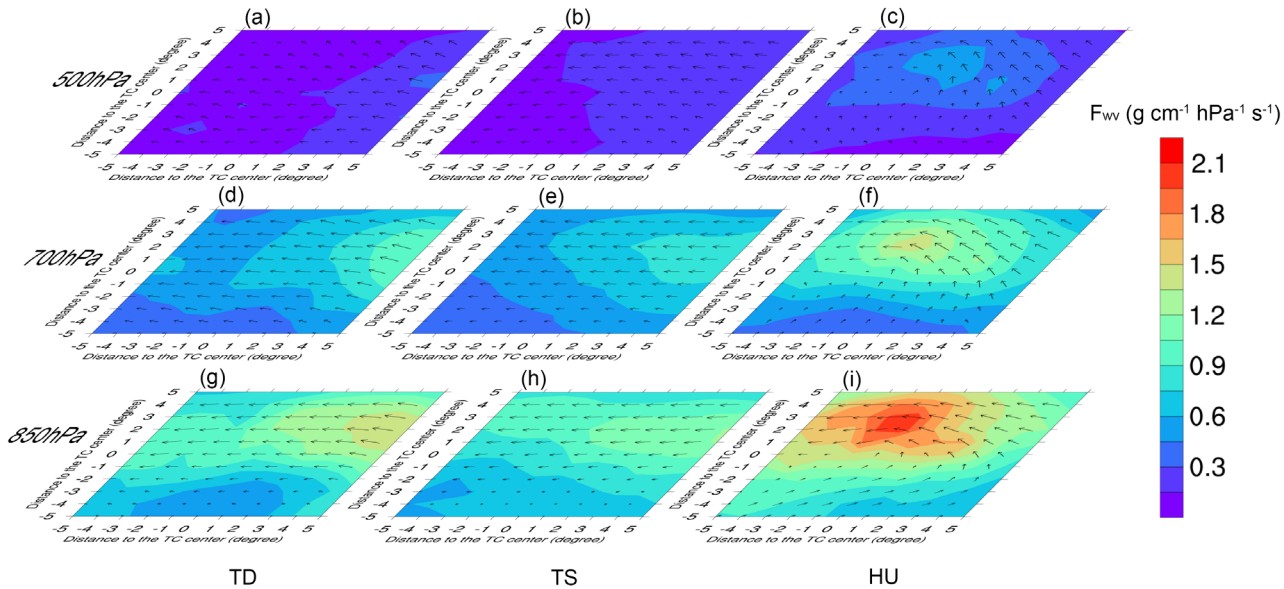

**Figure 6: Water vapor fluxes and wind fields on different isobaric surfaces of TD, TS, and HU, respectively: (a-c) 500 hPa, (d-f)**
**700 hPa, and (g-i) 850 hPa.**

Fig. 7 shows the vertical profiles of temperature and dew point, and they are the indicators of the thermal and moisture conditions. The temperature difference between pristine and polluted conditions is not obvious, and there is only a little heating when the SAL outbreaks. The variations of the dew point profile are significantly different by the intensities of TC. The dew point depression is the difference between the temperature and dew point at a certain height. For a constant
temperature, the smaller the difference, the more moisture there is, and the higher the relative humidity. In comparison to pristine conditions, the profiles of TD and TS show that relative humidity is higher at 700-450 hPa but lower at 400-250 hPa under polluted conditions. However, when the SAL occurs, the relative humidity of HU further increases at the height from 400 hPa to 200 hPa. Humidity is closely associated with the TC intensity (Juračić and Raymond, 2016), so it is clear that the SAL has negative effects on the TD and TS by evaporating and sinking the nephsystems but inversely invigorates the HU.

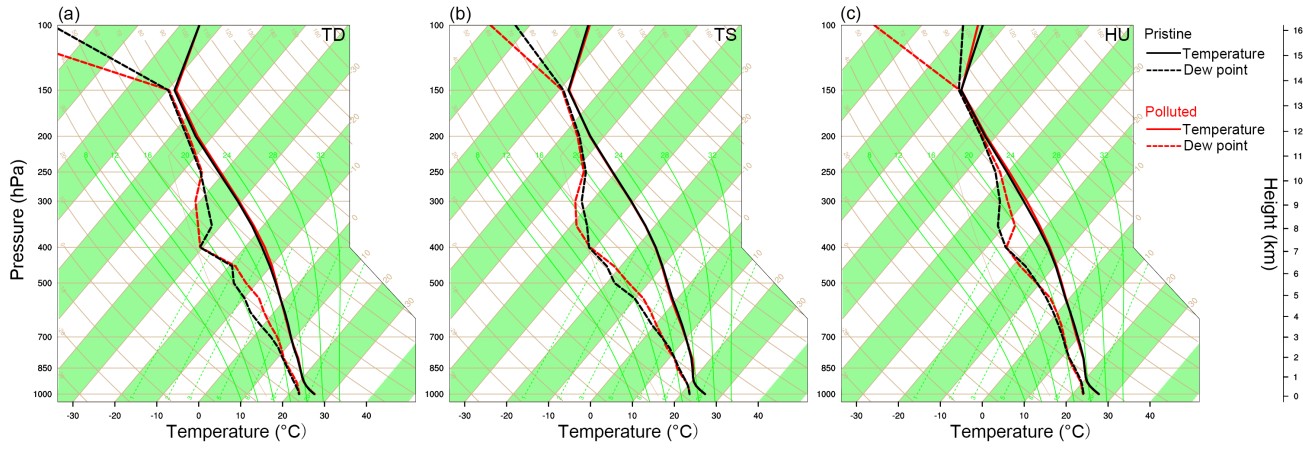

**Figure 7: SkewT-logP diagrams under pristine and polluted conditions by different TC intensities: (a) TD, (b) TS, and (c) HU.**

The interaction between TC and SAL is affected by multiple factors, which is the reason for the dew point profile differences of each TC intensity shown in Fig. 7. The combined influences of the thermal effect of dry SAL air, the microphysical effect of dust aerosols, and the dynamic effect of TCs are the reasons for the various responses of the TCs to SAL. To confirm the combined influences on the TCs, we further investigate the detailed feedback mechanisms of TCs to the SAL in the next section.

### 3.3 Impacts of the SAL on the TCs

The macroscopic and microphysical characteristics of the TCs with different intensities vary significantly, and their feedback mechanisms to the SAL are different due to their different dynamic and thermodynamic conditions. Therefore, it is necessary to distinguish them when discussing the effects of SAL on the TCs. The details are discussed from macro and micro perspectives respectively in the following two subsections.

### 3.3.1 Macro properties of TCs

The influences of the SAL on the CTT are compared among three different TC intensities displayed in Fig. 8. The CTTs of TD and TS have similar variations and tend to be warmer when the SAL outbreaks, and the trend of the warming CTT of TD is more significant than that of TS. It demonstrates that the negative effects of SAL on the TD and TS are related to the TC intensity, and weak TC is more likely to be destroyed by the SAL. However, when the environment is dust polluted, the CTT of the HU will be colder, indicating a positive response of HU to the SAL. Within a radius of about 350 km from the TC center, the CTT of the dusty HU reaches 235 K or lower, which is the temperature required for homogeneous nucleation.

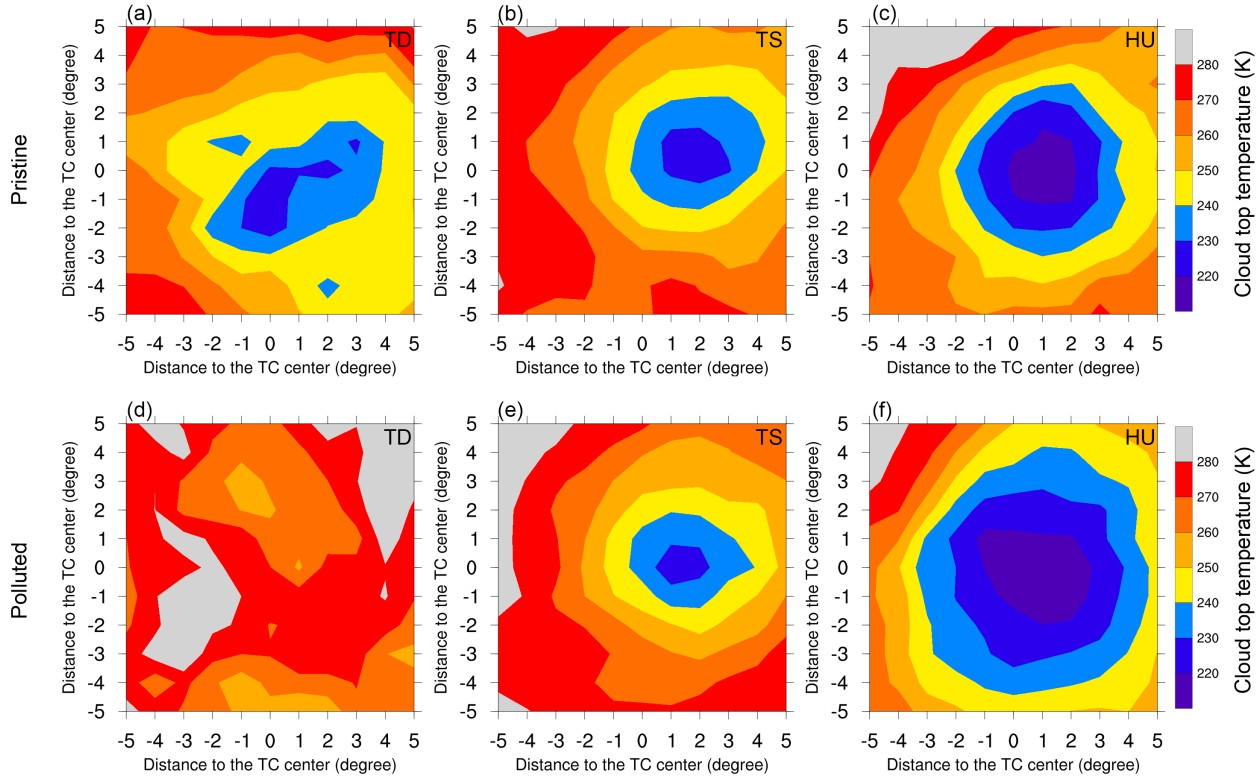

**Figure 8: The spatial distributions of CTT under (a-c) pristine and (d-e) polluted conditions of the composite TD, TS, and HU, respectively.**

By comparing the radial distributions of CTT with or without SAL loadings shown in Fig. 9, it can be seen that the responses of TCCS to SAL are suppressed (TD and TS) or enhanced (HU) both vertically and horizontally. The SAL is not limited to altering the thickness of the clouds, but also affects the horizontal range of the TCs, especially the size of the anvils. Koren et al. (2014), Yan et al. (2014) and Zhao et al. (2018) also obtained the same conclusion that aerosols both affect convective cloud thickness and horizontal range. The upper anticyclone formed by divergence restrains the development of convection, while TD is less disturbed by anticyclone compared with TS. When a number of convective cells develop independently around the TD center, the CTT of each convective cell can reach a lower temperature. Therefore, the CTT of TD is colder than that of TS in pristine conditions.

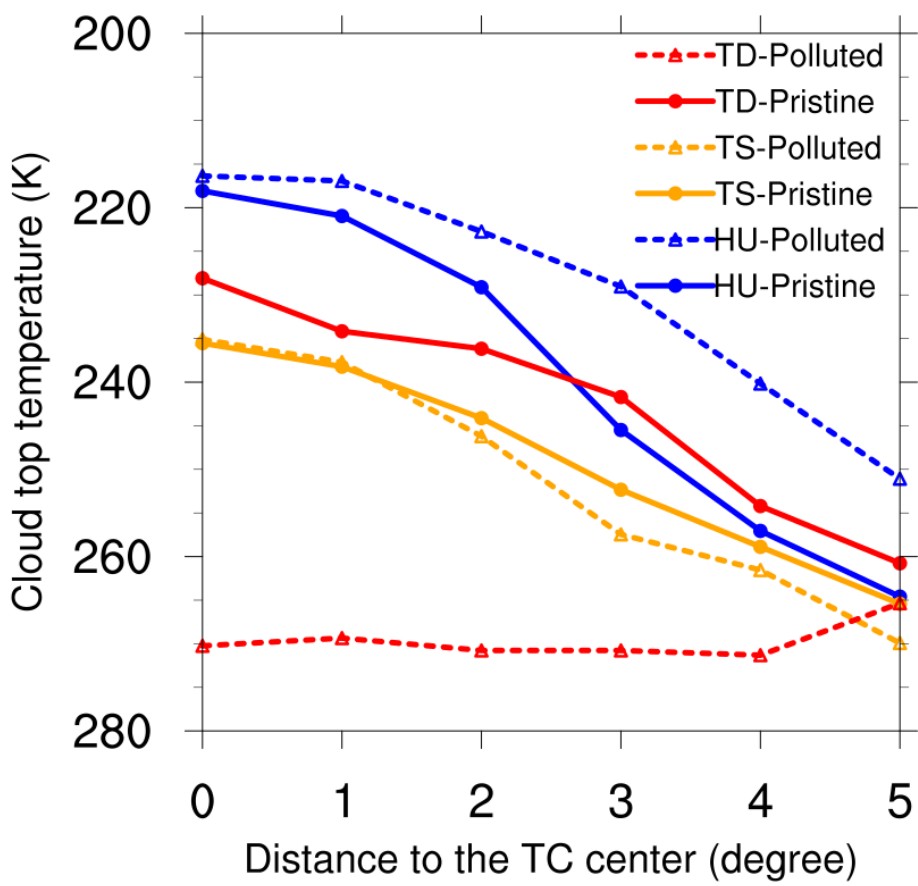

**Figure 9: The radial distributions of CTT under pristine and polluted conditions of the composite TD, TS, and HU, respectively.**

The competition between the updraft of the TC and the evaporation by SAL determines the variation of the CTT. The SAL may inhibit the TC intensification in two ways: on the one hand, the entrainment of dry and stable air into storms promotes convectively driven downdrafts in TCs (Fritz and Wang, 2013); on the other hand, as absorbing aerosols, mineral dust in the SAL absorbs solar radiation to heat the atmosphere, enhancing cloud evaporation and stabilizing the environment, known as the semi-direct effect (Helmert et al., 2007). For TD and TS, the updrafts of the TCs are relatively low and the evaporations by the SAL play the leading role. The increase of SAL intensity evaporates the ice cloud and then warms the CTT. But for HU, the violent updraft transports large amounts of water vapors from the ocean to prevent the inhibition of the TCCS influenced by the SAL. In addition, large amounts of dust aerosols transport to the high altitude along with the strong updraft will increase the concentrations of ice nuclei. More opportunities for heterogeneous freezing will enhance the released latent heat, and further invigorate the vertical development of HU. This is so-called the aerosol invigoration effect (Andreae et al., 2004; Koren et al., 2005). Therefore, the CTT of HU is much colder when influenced by the SAL.

### 3.3.2 Micro properties of TCs

The statistical distributions of the SAL-influenced cloud microphysical properties under different TC intensities are compared and evaluated in terms of PDFs. Fig. 10 denotes the PDFs of $CER_i$ under pristine and polluted conditions for TD, TS and HU. The PDFs of $CER_i$ show a range between 5 and 60 μm, and the peaks are around 30-35 μm. The results indicate that the impacts of SAL on the $CER_i$ exhibit different responses to the TC intensity. It is noting that the impact of SAL on the variation of $CER_i$ is insignificant for TS, whereas the PDFs of $CER_i$ have differentiated changes for TD and HU. When

the SAL outbreaks, the PDF curve of $CER_i$ generally shifts to the smaller value for TD (the peak $CER_i$ decreases from 33.76 to 32.11 μm), whereas the curve becomes flatter for HU (the peak $CER_i$ remains constant, while the PDF kurtosis decreases).

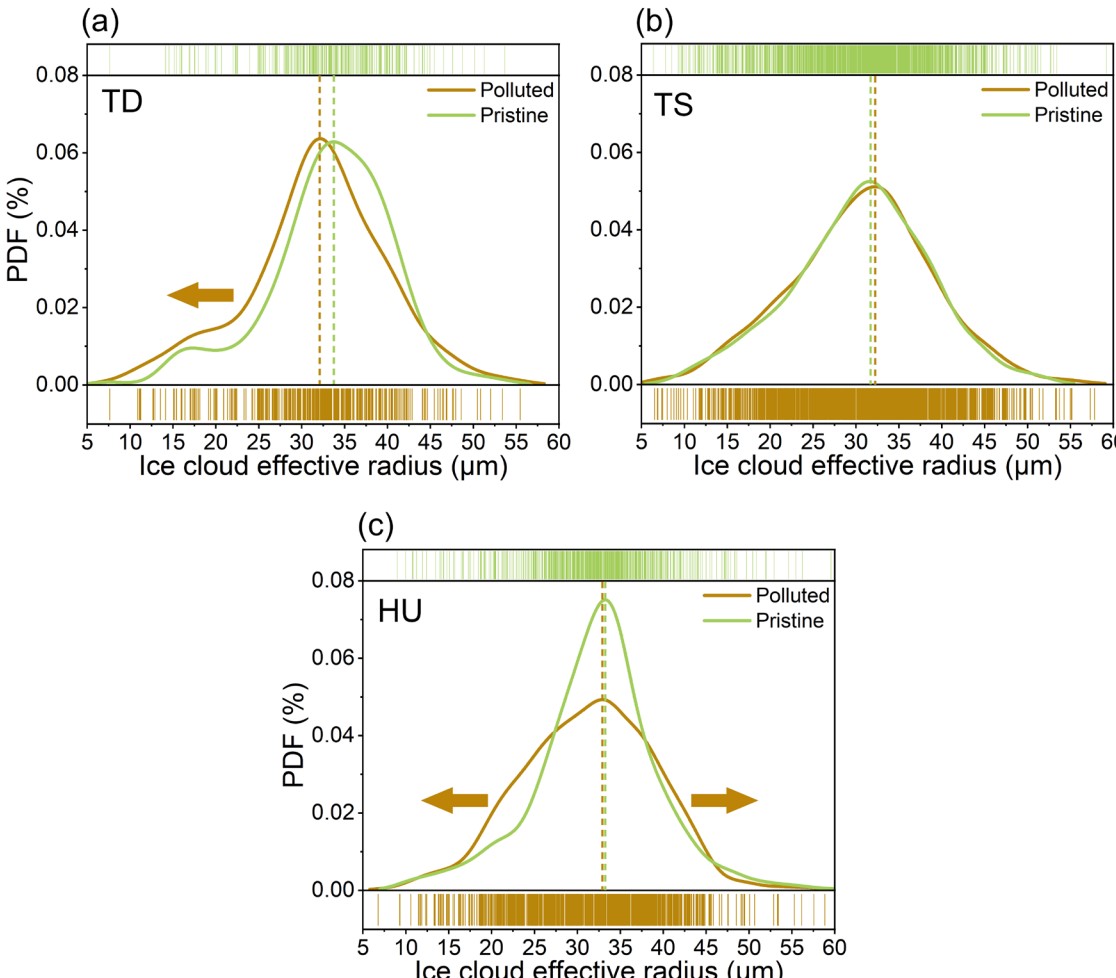

**Figure 10: The PDFs of CERᵢ under pristine and polluted conditions by different TC intensities: (a) TD, (b) TS, and (c) HU. The dashed lines indicate the peak perpendicular lines of the PDFs.**

CER$_i$ is related to both CIWP and CTT, and the larger size of ice particles indicates greater CIWP or warmer CTT (Wyser, 1998). The dry and dusty SAL influences the ice cloud particle size in serval ways. On the one hand, from the macro perspective, when the SAL inhibits the TCs, the CTT warms, and CIWP decreases. Meanwhile, the warm CTT leads to an increase of CER$_i$, and the reduced CIWP decreases CER$_i$. On the other hand, dust aerosols serve as IN, which decreases CER$_i$ and increases CIWP through the microphysical effect. The macro and micro effects cause the opposite impact on

CIWP, so it is an effective way to estimate which of the two effects play a more significant role by the variation in CIWP.

As demonstrated in Fig. 11, the CTTs of TD and TS both decrease when influenced by SAL due to the evaporation effect, and it is more remarkable for TD. However, the CIWPs of these two TC classifications show different variation patterns. When the SAL outbreaks, the CIWP of TD decreases, but the CIWP of TS remains constant. This result denotes that the SAL primarily decreases the CER$_i$ for TD due to the smaller CIWP when evaporated by the macro effect. However,

the response of CER$_i$ to SAL is insignificant for TS due to the balance between warming CTT (increasing CER$_i$) and dust indirect effect (decreasing CER$_i$). The difference in the impacts of SAL on CER$_i$ between TD and TS is responsible for their different intensities expressed in Fig. 5. Weaker TCs are more susceptible to be evaporated by dry air masses due to their insufficient water vapor supply.

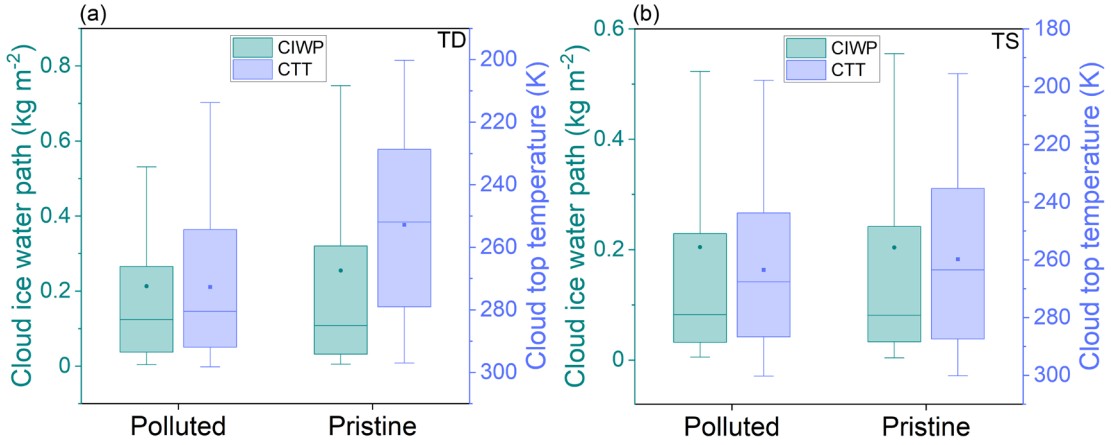

**Figure 11: Box plots of CIWP and CTT under polluted and pristine conditions by different TC intensities: (a) TD, and (b) TS. The dots and squares represent the average values of CIWP and CTT, respectively. The horizontal lines inside the boxes are the medians, and the bottom and top sides of the boxes represent the first and third quartiles. The whiskers are the minimum (maximum) values within 1.5 interquartile ranges of the lower (upper) quartile.**

The water vapor supply of HU is the most sufficient when compared with the other two TC classifications, so the

evaporation by the SAL no longer produces a marked effect. Given the different PDF variations in different CER$_i$ intervals of HU, we sort out three intervals of CER$_i$ (18-25 μm, 30-35 μm, 38-45 μm) corresponding to different patterns of variability. When analyzing the interval of CERi between 38 and 45 μm, it is clear that the more proportion of CER$_i$ is caused by dust aerosols nucleations within the height when the CTT ranges from 205 to 230 K shown in the shadow area in Fig. 12c. The dust aerosols are carried to the high altitude by the violent updraft, and the CER$_i$ produced by heterogeneous nucleation in

the polluted condition is larger than that by homogeneous nucleation in the pristine condition. In the interval of CER$_i$ between 18 and 25 μm, the more proportion of CER$_i$ within the height when the CTT ranges from 200 to 220 K is caused by the aerosol invigoration effect in the polluted conditions (Fig. 12a). The latent heat released by heterogeneous nucleation with more ice nuclei in the low layer further invigorates convection, inducing the colder CTT and the smaller CER$_i$. When the proportions of smaller and larger CER$_i$ increase simultaneously, the density of CER$_i$ between 30 and 35 μm decrease

naturally. Therefore, combined with the above reasons, the PDF curve of HU presented in Fig. 10c becomes flat when the cloud is influenced by SAL.

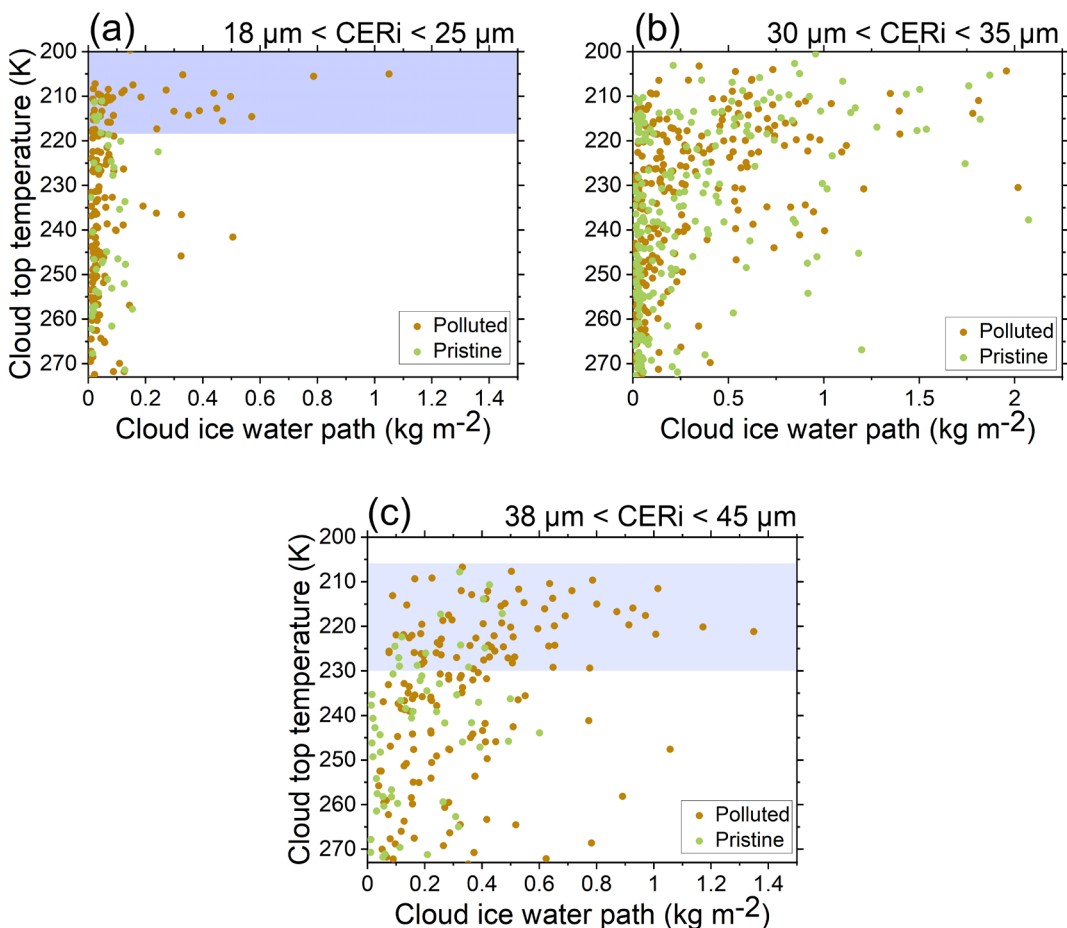

**Figure 12: Scatter plots between CIWP and CTT of HU under different conditions of CER$_i$: (a) 18 μm < CER$_i$ < 25 μm, (b) 30 μm < CER$_i$ < 35 μm, and (c) 38 μm < CER$_i$ < 45 μm.**

**4 Conclusions and Summary**

Using 17 years (from 2003 to 2019) of co-located cloud and aerosol observations from MODIS C6.1 MYD08-D3 products, together with Atlantic TC track data from NOAA's Tropical Prediction Center and NCEP-FNL global reanalysis meteorological dataset, we have investigated the responses of TCs to SAL from macro and micro views. The SAL over the tropical Atlantic can affect TC activities by impacting the atmospheric hydrology and thermodynamic structure. The distinct effects of SAL on the different intensities of TCs have been separated and examined, and the reasons have been discussed from the perspective of combined influences of dust aerosols, dry air masses, and TC updrafts, respectively.

Based on the 70 TC samples from July to September in the years 2003-2019, the composite spatial distributions of different TC intensities (TD, TS and HU) have been analyzed. The results indicate that the compactness of the nephsystem is related to the TC intensity. The structure of the TD nephsystem is loose, whereas the center of HU is noticeable and the structure of HU is highly symmetrical and well organized. Besides, the sufficient temporal and spatial mixing of SAL and TCCS has been demonstrated. The SAL impacts TCCS hydrology by the entrainment of dry and dusty air masses, thus modifying the TC activity remarkably.

The macro and micro characteristics of the TCCS are analyzed in this study by the variability of CTT and $CER_i$, respectively. The influence mechanism of SAL on TCCS is further visualized in Fig. 13 on the basis of 17-year observations. When affected by the SAL, the CTTs of weak TCs (TD and TS) are warmed while the colder CTT of HU is manifest (Fig. 13). The SAL is not limited to altering the thickness of the nephsystem, but also affects the horizontal range of the TCCS, especially the size of the anvils. Our analysis shows that the impact of SAL on the variation of $CER_i$ for TS is insignificant, whereas the PDFs of $CER_i$ have distinct changes for TD and HU. When affected by SAL, the PDF curve of $CER_i$ generally shifts to the smaller value for TD, whereas the curve becomes flatter for HU (shifting to both smaller and larger values).

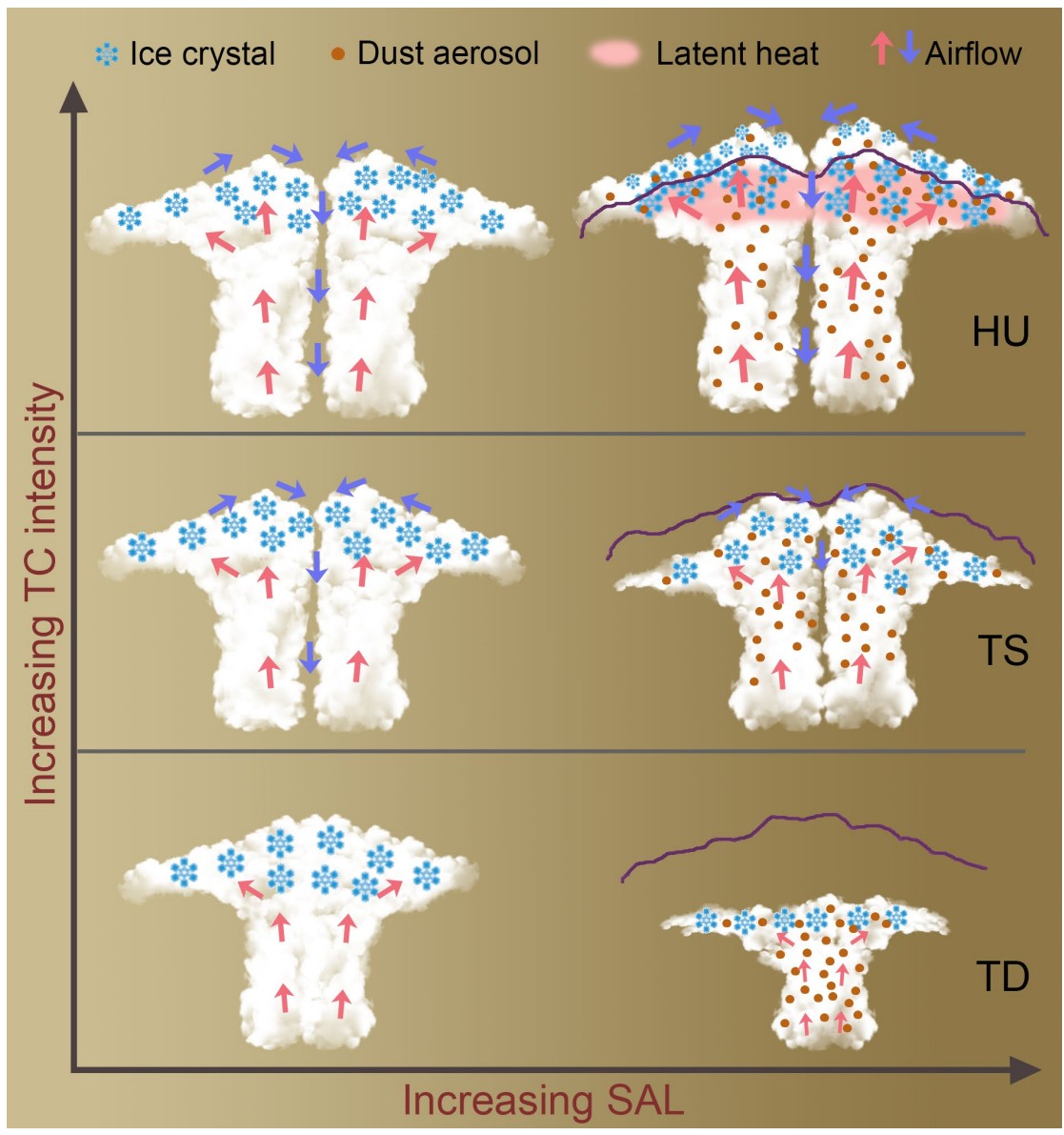

**Figure 13: The scheme description of the effects of SAL on the TCCS. The gradual desert color backdrop indicates the intensity of SAL. The purple curves are the cloud top heights under pristine conditions.**

The various feedback mechanisms of TCCS to SAL are determined by the combined effects of SAL dry air masses, the dust aerosols as ice nuclei, as well as the dynamic, thermodynamic and moisture conditions. The specific influence mechanisms of SAL on the three types of TC (TD, TS and HU) are explored, and the physical implications are pointed out here:

(a) TD: Dry air masses in the SAL and the absorbed radiation by the mineral dust aerosols evaporate the TD nephsystem, leading to the systematic decreasing of cloud top height. Due to the weak updraft and lack of moisture, plenty of dust aerosols as ice nuclei compete for the fixed water vapor, which reduces the size of ice particles.

(b) TS: The evaporation effects by the SAL contribute to the decreasing height of the spiral cloud band. However, the response of $CER_i$ to SAL is insignificant with the decreasing CTT, which is due to the balance between warming CTT and dust indirect effect, that is, the warming CTT increases the $CER_i$ while the dust indirect effect reduces the cloud particle size.

    (c) HU: The horizontal range and thickness of HU are both strengthened by the impact of SAL, which is contributed by the dust aerosol invigoration effect. The violent updraft transports large amounts of water vapors from the Atlantic to

325 prevent the evaporation of the ice cloud influenced by the SAL, so the evaporation by the SAL no longer produces a marked effect. The effects of SAL on HU are divided into two aspects. On the one hand, large amounts of dust aerosols transport to the high altitude along with the strong updraft increase the concentrations of ice nuclei, and the $CER_i$ produced by heterogeneous nucleation in the polluted condition is larger than that by homogeneous nucleation in the pristine condition. On the other hand, the more opportunities for heterogeneous freezing enhance the released latent heat, and further invigorate

the HU, inducing the higher cloud top and the smaller $CER_i$.

    The results presented in this study provide evidence to investigate the physical mechanisms between the SAL and TCCS using long periods of detailed satellite measurements. The conclusions are somewhat beneficial to our recognition and forecast of the physical processes and evolutions of TCs in the Atlantic region. However, it is quite difficult to evaluate these uncertainties in this study due to the limitations of the data, and future work will explore the physical implications with aid

of in-situ observations and numerical simulations.

**Code/Data availability**

All datasets used here are publicly available. The aerosol and cloud data taken from the MODIS-Aqua MYD08-D3 daily gridded products are available from https://search.earthdata.nasa.gov. Atlantic TC track data are obtained from the NOAA's Tropical Prediction Center (https://www.nhc.noaa.gov/data/#hurdat). The meteorological data taken from the NCEP-FNL

global reanalysis dataset are available from https://rda.ucar.edu/datasets/ds083.2. The codes that support the findings of this study are available from the corresponding author upon reasonable request.

**Author Contributions**

Y.H. designed research and edited the paper; H. L., Y. H. performed research and analyzed data; H. L. wrote the manuscript.

**Competing Interests**

The authors declare they have no conflict of interest.

**Acknowledgements**

This work was jointly supported by the National Natural Science Foundation of China (Grant Nos. 41775026, 42027804, 41075012 and 40805006). We thank the two anonymous reviewers for their constructive comments that have helped us to improve the manuscript.

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
