# Peer review of "Impacts of the Saharan air layer on the physical properties of the Atlantic tropical cyclone cloud systems: 2003-2019"

_Atmospheric Chemistry and Physics, 2021_

## Author Comment (AC1)

**Responses to comments of "Impacts of the Saharan air layer on the physical properties of the Atlantic tropical cyclone cloud systems: 2003–2019 [Preprint acp-2021-462]" to** *Atmospheric Chemistry and Physics***.**

Hao Luo, Yong Han

We would like to thank the editor Dr. Timothy J. Dunkerton and the reviewers for giving constructive criticisms and comments, which are very helpful in improving the quality of the manuscript. We have made the point-by-point response to the comments below and revised the manuscript accordingly. We hope that the revised version can meet the favorable approval and journal requirements. The referee's comments are reproduced (*black, italic*) along with our replies (blue) and changes made to the text (red) in the revised manuscript. All the authors have read the revised manuscript and agreed with the submission in its revised form. Please check them.

**Responses to Reviewers**

**Anonymous Referee #1**

*Luo and Han "Impacts of the Saharan air layer on the physical properties of the Atlantic tropical cyclone cloud systems: 2003-2019"*

**General comments:**

*This study investigates the impacts of the Saharan air layer (SAL) on the physical properties of the Atlantic tropical cyclone cloud systems (TCCS) for the summer of 17 years. It divided the 70 TC samples into three categories, tropical depression (TD), tropical storm (TS), and hurricane (HU). It found that the SAL makes crucial impacts on the generation and evolution of Tropical cyclones (TCs). The authors have attempted to identify whether and how the effects of the SAL play a positive or negative role on the TCCS, and to draw a qualitative conclusion of how SAL affects the various intensities of the TCs. The conclusions are beneficial to better understanding of the physical processes and evolutions of TCs in the Atlantic region. In addition, the manuscript is well written. I would recommend its acceptance for publication after necessary revisions.*

**Response:** We would like to thank you for your time in reviewing this manuscript. We sincerely appreciate your positive comments and valuable suggestions. We have revised the manuscript according to your constructive suggestions, which helps improve the quality of this paper.

**Comment NO.1:**

*Considering that this article involves lots of satellite and ground-based data, such as AOD, CTT, CERi, and CIWP, Atlantic TC track data and meteorological data etc., I would suggest that the authors add more information about the data matching in part 2.*

**Response:** Thank you for your valuable suggestion. We have added the data matching method in the revised version.

**Changes in Manuscript:**

"2.4 Methodology for data matching

Considering that this work involves multiple satellite and reanalysis data, such as AOD, CTT, CERi, and CIWP, Atlantic TC track data, meteorological data, etc., we provide more information about the data matching here. The specific study area is different for each TC, and each TC study area is located within $\pm 5$ degrees latitude or longitude away from the TC center, which is a grid of 11°×11°. The spatial resolution of all the data are 1°×1°, which allows the data to match well in the space. The time interval among the datasets is within 1.5 hours to guarantee a less changed atmospheric state. The CTT, CER$_i$ and CIWP in the study area are used to analyze the spatial TCCS properties. The AOD in the study area is averaged to characterize the SAL intensity. It should be noted that AOD is detected in the absence of clouds, so the AOD data are distributed on the periphery of the 11°×11° grid. Nevertheless, the properties of background aerosols are capable to be characterized by the average AOD. The meteorological data (T, Td, WS, WD, RH, etc.) in the study area are applied to examine the meteorological fields and profiles." **[Page 6 Lines 134-143 (in the "Track**

**Changes" version)]**

**Comment NO.2:**

*A schematic diagram in Part 3 to reflect the relationship between 3.1-3.3 might be helpful to the readers.*

**Response:** Thank you for your insightful suggestion. We have added a schematic diagram that reflects the relationship between sections 3.1-3.3 in the revision.

**Changes in Manuscript:**

"The relationship between sections 3.1-3.3 is demonstrated in Fig. 3." **[Page 6 Line 150 (in the "Track Changes" version)]**

[Figure]

**Figure 3: The schematic diagram reflecting the relationship between sections 3.1-3.3.**

**Comment NO.3:**

*Line 33, I am not sure if this description is accurate or not. The North Atlantic is one of the areas with the most frequent TCs in the world, along with the west pacific. However, I am not sure if the North Atlantic is an area with the most frequent TCs in the world. Actually, in my understanding the west Pacific have more and stronger TC*

*activities than the North Atlantic. Anyway, please provide a reference to support your claim here.*

**Response:** Thank you for your valuable comment and kind suggestion. According to the TC records by year (https://en.wikipedia.org/wiki/Tropical_cyclones_by_year), the Western Pacific is an area with the most frequent TCs in most years as you mentioned, but the records of 2020 reveal that TCs in the Atlantic region are the most frequent. As you suggested, to avoid mistakes, we have rewritten this sentence and indicate that the North Atlantic is one of the areas with the most frequent TCs in the world.

**Changes in Manuscript:**

"The North Atlantic is one of the areas with the most frequent TCs in the world, and these TCCS are often affected by the Saharan air layer (SAL), which is an elevated layer of Saharan dry air and mineral dust, especially during the boreal summer between July and September (Dunion and Velden, 2004)." **[Page 2 Lines 34-36 (in the "Track Changes" version)]**

**Comment NO.4:**

*Line 35-37, regarding these three kinds of aerosol radiative effects, three more references are recommended here, Albrecht (1989, doi: 10.1126/science.245.4923.1227), Garrett and Zhao (2006, doi: 10.1038/nature04636) and Zhao and Garrett (2015, doi: 10.1002/2014GL062015).*

**Response:** Thank you very much for your suggestion. We have cited the related references as you suggested, which provide more scientific support for this study.

**Changes in Manuscript:**

"Saharan dust usually propagates downstream along with SAL to the Atlantic region, which can modify the Earth-atmosphere system energy budget through direct, semi-direct and indirect radiative forcing, respectively (Twomey, 1974; Carlson and Benjamin, 1980; Albrecht, 1989; Sassen et al., 2003; Garrett and Zhao, 2006; Zhao and Garrett, 2015; Luo et al., 2019)." **[Page 2 Lines 36-39 (in the "Track Changes" version)]**

**Added references:**

Albrecht, B. A.: Aerosols, Cloud Microphysics, and Fractional Cloudiness, Science, 245, 1227, 10.1126/science.245.4923.1227, 1989.

Garrett, T. J., and Zhao, C.: Increased Arctic cloud longwave emissivity associated with pollution from mid-latitudes, Nature, 440, 787-789, 10.1038/nature04636, 2006.

Zhao, C., and Garrett, T. J.: Effects of Arctic haze on surface cloud radiative forcing, Geophysical Research Letters, 42, 557-564, 10.1002/2014GL062015, 2015.

**Comment NO.5:**

*Line 40-45: In my understanding, mineral dust particles have weak absorption and strong scattering, therefore, it is suggested that the author to examine the statement of this sentence "As absorbing aerosols, mineral dust particles absorb solar radiation and heat the atmosphere through the semi-direct effect, further enhancing cloud evaporation".*

**Response:** Thank you for your insightful suggestion. We have rephrased this sentence.

**Changes in Manuscript:**

"In addition, mineral dust particles heat the atmosphere and further enhance cloud

evaporation, referred to as the semi-direct effect." **[Page 2 Lines 42-44 (in the "Track Changes" version)]**

**Comment NO.6:**

*Line 50-58, The recent study by Sun and Zhao (2020) have done a comprehensive analysis about the dust aerosol effects on the meteorological environment, including the vorticity, temperature, wind shear, and humidity, which suggested a negative effect to TC formation. This recent study is worthy to read and mention here.*

**Response:** Thank you for your valuable comment. This work has been mentioned in the revision, which provides more scientific support for our study.

**Changes in Manuscript:**

"Sun and Zhao (2020) have conducted a comprehensive investigation into the effects of dust aerosols on the meteorological environment, including the vorticity, temperature, wind shear, and humidity, which indicated a negative effect on TC formation." **[Pages 2-3 Lines 62-64 (in the "Track Changes" version)]**

**Comment NO.7:**

*Line 62-64: Add a word "all" before the factors.*

**Response:** Done.

**Changes in Manuscript:**

"Note that the meteorological parameters include sea surface temperature, vorticity, wind shear, specific humidity, and planetary boundary layer are all the factors for the formation and development of TCs" **[Page 3 Lines 69-71 (in the "Track Changes" version)]**

**Comment NO.8:**

*Line 94-95: Add a word "etc." at the end of this sentence.*

**Response:** Done.

**Changes in Manuscript:**

"The data include the AOD, CTT, $CER_i$, CIWP, etc." **[Page 4 Lines 106-107 (in the "Track Changes" version)]**

**Comment NO.9:**

*Figure 1, it should be noted that the aerosol property (AOD) is generally averaged for time when there are no clouds (so no TC activities).*

**Response:** Thank you for your valuable suggestion. We have noted it in the caption of Figure 1.

**Changes in Manuscript:**

"It should be noted that the AOD is generally averaged in the absence of clouds, i.e. no TC activities." **[Page 4 Line 102 (in the "Track Changes" version)]**

**Comment NO.10:**

*Line 113-116, The question is how accurate the meteorological variables from FNL reanalysis data are.*

**Response:** Thank you for your critical comment. We have clarified the accurate and wide applications of FNL reanalysis data in the revision.

**Changes in Manuscript:**

"The FNL reanalysis datasets incorporate additional observations that are not available for inclusion in the real-time NCEP Global Forecast System (GFS) analysis, which have

been widely applied to many studies (Ritchie et al., 2011; Kerns and Chen, 2013)."

**[Pages 5-6 Lines 129-132 (in the "Track Changes" version)]**

**Comment NO.11:**

*Line 155: Add "respectively" at the end of this sentence.*

**Response:** Done.

**Changes in Manuscript:**

"Fig. 6 depicts the water vapor flux and wind field on the isobaric surfaces of 500 hPa, 700 hPa, and 850 hPa, respectively." **[Page 9 Lines 184-185 (in the "Track Changes" version)]**

**Comment NO.12:**

*Line 168-170: Rephrase this sentence.*

**Response:** Thank you. We have rephrased this sentence.

**Changes in Manuscript:**

"In comparison to pristine conditions, the profiles of TD and TS show that relative humidity is higher at 700-450 hPa but lower at 400-250 hPa under polluted conditions."

**[Page 10 Lines 199-200 (in the "Track Changes" version)]**

**Comment NO.13:**

*Line 200-201, Similar findings have also been found by a recent study of Zhao et al. (2018, doi: 10.1029/2018GL079427), which showed that aerosols can cause broader precipitation area (horizontal range of TC) and weaker maximum precipitation intensity (lower cloud thickness at the band near TC eye).*

**Response:** Thank you for your comment. The conclusion of Zhao et al. (2018) is critical

to the literature support of this study, which has been cited in this paper.

**Changes in Manuscript:**

"Koren et al. (2014), Yan et al. (2014) and Zhao et al. (2018) also obtained the same conclusion that aerosols both affect convective cloud thickness and horizontal range."

**[Page 12 Lines 230-231 (in the "Track Changes" version)]**

**Added reference:**

Zhao, C., Lin, Y., Wu, F., Wang, Y., Li, Z., Rosenfeld, D., and Wang, Y.: Enlarging Rainfall Area of Tropical Cyclones by Atmospheric Aerosols, Geophysical Research Letters, 45, 8604-8611, 10.1029/2018GL079427, 2018.

**Comment NO.14:**

*Figure 9, using "μm" might be better for the cloud effective radius unit.*

**Response:** Thank you. We have replaced the unit "micron" with "μm" in Figure 9 (Figure 10 in the revision).

**Changes in Manuscript:**

[Figure]

**Figure 10: The PDFs of CER$_i$ under pristine and polluted conditions by different TC intensities: (a) TD, (b) TS, and (c) HU. The dashed lines indicate the peak perpendicular lines of the PDFs. [Original Figure 9]**

**Comment NO.15:**

*Line 220-225: Adding the peak perpendicular line of the probability density function to FIG. 9, clarifying the moving value of the peak, the specific situation can be clearly marked on FIG. 9.*

**Response:** Thank you for your constructive suggestion. The peak perpendicular lines of the PDF curves have been added in Fig. 9 (Fig. 10 in the revision). Moreover, the moving value of the peak has been clarified in the revision.

**Changes in Manuscript:**

"When the SAL outbreaks, the PDF curve of CER$_i$ generally shifts to the smaller value

for TD (the peak $CER_i$ decreases from 33.76 to 32.11 µm), whereas the curve becomes flatter for HU (the peak $CER_i$ remains constant, while the PDF kurtosis decreases)."

**[Page 14 Lines 253-256 (in the "Track Changes" version)]**

**Comment NO.16:**

*Line 267-269: Add "respectively" at the end of this sentence.*

**Response:** Done.

**Changes in Manuscript:**

"The distinct effects of SAL on the different intensities of TCs have been separated and examined, and the reasons have been discussed from the perspective of combined influences of dust aerosols, dry air masses, and TC updrafts, respectively." **[Page 17 Lines 300-302 (in the "Track Changes" version)]**

---

## Author Comment (AC2)

**Responses to comments of "Impacts of the Saharan air layer on the physical properties of the Atlantic tropical cyclone cloud systems: 2003–2019 [Preprint acp-2021-462]" to** *Atmospheric Chemistry and Physics***.**

Hao Luo, Yong Han

We would like to thank the editor Dr. Timothy J. Dunkerton and the reviewers for giving constructive criticisms and comments, which are very helpful in improving the quality of the manuscript. We have made the point-by-point response to the comments below and revised the manuscript accordingly. We hope that the revised version can meet the favorable approval and journal requirements. The referee's comments are reproduced (*black, italic*) along with our replies (blue) and changes made to the text (red) in the revised manuscript. All the authors have read the revised manuscript and agreed with the submission in its revised form. Please check them.

**Responses to Reviewers**

**Anonymous Referee #2**

*Review of "Impacts of the Saharan air layer on the physical properties of the Atlantic tropical cyclone cloud systems: 2003-2019" by Luo and Han.*

**General comments:**

*This is an interesting study that examines the impacts of the Saharan air layer (SAL) on the physical features of the Atlantic tropical cyclone (TC) cloud systems (TCCS) by using co-located satellite-based cloud and aerosol observations. The authors attempt to distinguish the various effects of SAL on different intensities of TCs, and further find out the probable causes of the varied feedback mechanisms. Based on the 70 TC samples during the summertime of 2003-2019, the varying impacts of SAL on different intensities of TCCS are well analyzed and concluded. They find that the various responses of TCCS to SAL are determined by the combined factors of dry air masses, the dust aerosols as ice nuclei, as well as the thermodynamic and moisture conditions. These conclusions may contribute to a better understanding of the physical mechanisms between the SAL and TCs over the Atlantic region. Overall, the manuscript is well written and the data are well analyzed and presented. Therefore, I would recommend its acceptance for publication in ACP after minor revisions.*

**Response:** We would like to thank you for your time in reviewing this manuscript. We sincerely appreciate your positive assessment of our manuscript. We have revised the manuscript according to your suggestions, which helps us improve the quality of this paper.

**Specific comments:**

**Comment NO.1:**

*Line 22: Please rephrase this sentence. "but the PDF curve becomes flatten for HU"*

*should be "whereas the PDF curve becomes flatter for HU".*

**Response:** Thank you. This sentence has been rephrased.

**Changes in Manuscript:**

"When affected by SAL, the probability density function (PDF) curve of $CER_i$ generally shifts to the smaller value for TD, whereas the PDF curve becomes flatter for HU." **[Page 1 Lines 22-23 (in the "Track Changes" version)]**

**Comment NO.2:**

*Line 60: Please specify what is meant by TCCS macro and micro properties. After reviewing all of the figures and the remainder of the paper, it seems that the macro properties include the cloud top temperature and the cloud ice water path, while the micro properties refer to the ice cloud effective radius. Since "cloud macro or micro properties" is a rather broad term that could be interpreted in many different ways, please state this clearly at least once.*

**Response:** Thank you for your constructive suggestions and sorry for the unclear statements. We have provided clear definitions of TCCS macro and micro characteristics when introducing our work in the revised version.

**Changes in Manuscript:**

"Here, the TCCS macro characteristics include the CTT and CIWP, while the TCCS micro characteristics refer to the $CER_i$." **[Page 3 Lines 79-80 (in the "Track Changes" version)]**

**Comment NO.3:**

*Line 76: "characteristics" should be "variation characteristics".*

**Response:** Thank you. We have added "variation" before "characteristics".

**Changes in Manuscript:**

"To illustrate the variation characteristics of Atlantic TCCS influenced by the SAL, we use the aerosol and cloud data from MODIS-Aqua, the TC track data from NOAA's Tropical Prediction Center, and the meteorological data from the National Centers for Environmental Prediction-Final Operational Global Analysis (NCEP-FNL) dataset."

**[Page 3 Lines 84-86 (in the "Track Changes" version)]**

**Comment NO.4:**

*Line 224: Related to Line 22.*

**Response:** Done.

**Changes in Manuscript:**

"When the SAL outbreaks, the PDF curve of $CER_i$ generally shifts to the smaller value for TD (the peak $CER_i$ decreases from 33.76 to 32.11 μm), whereas the curve becomes flatter for HU (the peak $CER_i$ remains constant, while the PDF kurtosis decreases)."

**[Page 14 Lines 253-256 (in the "Track Changes" version)]**

**Comment NO.5:**

*Figure 9: The units of cloud effective radius should be "μm" to match the units shown in Figure 11.*

**Response:** Thank you. We have replaced the unit "micron" with "μm" in Figure 9 (Figure 10 in the revision).

**Changes in Manuscript:**

[Figure]

**Figure 10: The PDFs of CERᵢ under pristine and polluted conditions by different TC intensities: (a) TD, (b) TS, and (c) HU. The dashed lines indicate the peak perpendicular lines of the PDFs. [Original Figure 9]**

**Comment NO.6:**

*Line 258: "flatten" should be "flat".*

**Response:** Done.

**Changes in Manuscript:**

"Therefore, combined with the above reasons, the PDF curve of HU presented in Fig. 8c becomes flat when the cloud is influenced by SAL." **[Page 16 Lines 291-292 (in the "Track Changes" version)]**

**Comment NO.7:**

*Line 276: The "TC nephsystems" should be "TCCS" to be consistent with the context.*

**Response:** Thank you. We have revised it.

**Changes in Manuscript:**

"The macro and micro characteristics of the TCCS are analyzed in this study by the variability of CTT and $CER_i$, respectively." **[Page 18 Lines 309-310 (in the "Track Changes" version)]**

**Comment NO.8:**

*Line 277: Add a comma before "respectively".*

**Response:** Done.

**Changes in Manuscript:**

"The macro and micro characteristics of the TCCS are analyzed in this study by the variability of CTT and $CER_i$, respectively." **[Page 18 Lines 309-310 (in the "Track Changes" version)]**

**Comment NO.9:**

*Line 282: Related to Line 22.*

**Response:** Done.

**Changes in Manuscript:**

"When affected by SAL, the PDF curve of $CER_i$ generally shifts to the smaller value for TD, whereas the curve becomes flatter for HU (shifting to both smaller and larger values)." **[Page 18 Lines 314-316 (in the "Track Changes" version)]**

**Comment NO.10:**

*Line 287: "various feedbacks" should be "various feedback mechanisms". Please modify it throughout the context.*

**Response:** Thank you. We have modified it throughout the context.

**Changes in Manuscript:**

"The various feedback mechanisms of TCCS to SAL are …" **[Page 19 Line 320 (in the "Track Changes" version)]**

"… and further find out the probable causes of the varied feedback mechanisms." **[Page 1 Lines 15-16 (in the "Track Changes" version)]**

"this study comprehensively examines the feedback mechanisms of …" **[Page 3 Lines 72-73 (in the "Track Changes" version)]**

"… we further investigate the detailed feedback mechanisms of …" **[Page 11 Line 209 (in the "Track Changes" version)]**

"… and their feedback mechanisms to the SAL are different …" **[Page 11 Lines 213 (in the "Track Changes" version)]**

**Comment NO.11:**

*Line 310: Please provide relevant search terms for the public data retrieval in the data availability statement.*

**Response:** Thank you for your valuable suggestion. We have provided the relevant

search terms for the public data retrieval in the data availability statement.

**Changes in Manuscript:**

"All datasets used here are publicly available. The aerosol and cloud data taken from the MODIS-Aqua MYD08-D3 daily gridded products are available from https://search.earthdata.nasa.gov. Atlantic TC track data are obtained from the NOAA's Tropical Prediction Center (https://www.nhc.noaa.gov/data/#hurdat). The meteorological data taken from the NCEP-FNL global reanalysis dataset are available from https://rda.ucar.edu/datasets/ds083.2. The codes that support the findings of this study are available from the corresponding author upon reasonable request." **[Page 20 Lines 344-348 (in the "Track Changes" version)]**